# Factors affecting template switch recombination associated with restarted DNA replication

Manisha Jalan[†], Judith Oehler, Carl A Morrow[‡], Fekret Osman, Matthew C Whitby*

Department of Biochemistry, University of Oxford, Oxford, United Kingdom

**Abstract** Homologous recombination helps ensure the timely completion of genome duplication by restarting collapsed replication forks. However, this beneficial function is not without risk as replication restarted by homologous recombination is prone to template switching (TS) that can generate deleterious genome rearrangements associated with diseases such as cancer. Previously we established an assay for studying TS in *Schizosaccharomyces pombe* (Nguyen et al., 2015). Here, we show that TS is detected up to 75 kb downstream of a collapsed replication fork and can be triggered by head-on collision between the restarted fork and RNA Polymerase III transcription. The Pif1 DNA helicase, Pfh1, promotes efficient restart and also suppresses TS. A further three conserved helicases (Fbh1, Rqh1 and Srs2) strongly suppress TS, but there is no change in TS frequency in cells lacking Fml1 or Mus81. We discuss how these factors likely influence TS.
DOI: https://doi.org/10.7554/eLife.41697.001

*For correspondence:
matthew.whitby@bioch.ox.ac.uk

Present address: [†]Department of Radiation Oncology, Memorial Sloan Kettering Cancer Center, New York, United States; [‡]Department of Oncology, Weatherall Institute of Molecular Medicine, University of Oxford, Oxford, United Kingdom

Competing interests: The authors declare that no competing interests exist.

## Introduction

Problems that arise during DNA replication are thought to be the root cause of much of the genome instability, including duplications, deletions and translocations of chromosomal segments, that is associated with cancer and other disease states (*Aguilera and García-Muse, 2013*; *Arlt et al., 2012*; *Macheret and Halazonetis, 2015*). One common problem that arises is accidental collision of the replication fork with proteins that are tightly bound to the DNA (*Lambert and Carr, 2013*). Such collisions can precipitate a process known as replication fork collapse in which the protein components of the fork (collectively known as the replisome) are thought to partially or completely dissociate from the DNA (*Cortez, 2015*). In some cases this process may also be associated with DNA breakage, detaching one arm of the replication fork and exposing a single duplex DNA end. Replication fork collapse, either with or without DNA breakage, causes a localized cessation of DNA synthesis that can be remedied through the intervention of homologous recombination (HR) proteins, which restart replication.

In eukaryotes, recombination-dependent replication (RDR) has been studied mostly in the budding yeast *Saccharomyces cerevisiae*, using experimental systems in which RDR is initiated from a site-specific DNA double-strand break (DSB) (*Anand et al., 2013*; *Kramara et al., 2018*; *Sakofsky and Malkova, 2017*). HR uses intact homologous DNA sequences as templates to repair DSBs and, if such sequences are available for only one side of the break, repair occurs via a RDR process termed break induced replication (BIR). In most cases, BIR has been studied outside of the context of S-phase, with a DSB, rather than a collapsed or broken replication fork, acting as its inducer. However, studies, using a site-specific single-strand DNA break to induce replication fork breakage, have shown that essentially the same BIR process can repair broken forks during S-phase (*Mayle et al., 2015*).

Similar to other HR-mediated repair processes, BIR involves the 5' to 3' nucleolytic resection of a duplex DNA end to provide a single-stranded DNA (ssDNA) tail for the Rad51 recombination protein to load on to (*Anand et al., 2013*). In yeast, this loading reaction depends on Rad52, which acts as a mediator between ssDNA coated with the ssDNA-binding protein RPA and Rad51 (*Krogh and Symington, 2004*). Once loaded, Rad51 performs a DNA homology search, followed by strand invasion of the intact DNA sequence that it finds, forming a displacement (D) loop. The D-loop acts as a platform for replication protein reassembly, which seemingly involves most of the main replisome components (*Lydeard et al., 2010*). However, BIR exhibits a number of features that distinguishes it from canonical replication initiated from a replication origin, including: a requirement for the non-essential DNA Polymerase ∂ subunit Pol32 (*Lydeard et al., 2007*); a dependence on the Pif1 DNA helicase to promote DNA synthesis (*Saini et al., 2013*; *Wilson et al., 2013*); progression via a migrating D-loop or bubble resulting in conservative rather than semi-conservative replication (*Saini et al., 2013*); a massively increased rate of mutagenesis (*Deem et al., 2011*); an association with clustered mutations (*Sakofsky et al., 2014*); and an increased frequency of template switching (TS) (*Smith et al., 2007*). Given that a BIR-like process has been documented in human cells (*Costantino et al., 2014*; *Roumelioti et al., 2016*), it is highly likely that its error-prone nature and propensity for TS causes many of the mutations and genome rearrangements that are associated with cancer and other diseases. It is therefore imperative that the factors that influence BIR's tendency to cause genomic change are identified, and their conservation of function in related RDR processes assessed, especially in the context of ongoing DNA replication during S-phase.

We, and others, have documented a related process to BIR in the fission yeast *Schizosaccharomyces pombe*, which is induced by the *RTS1* protein-DNA polar replication fork barrier (RFB), as opposed to a DNA break (*Lambert et al., 2010*; *Miyabe et al., 2015*; *Mizuno et al., 2013*; *Nguyen et al., 2015*). This form of RDR has been termed homologous recombination-restarted replication (HoRReR) (*Miyabe et al., 2015*) but, in this paper, we will refer to it simply as RDR. Replication forks collapse, and recruit HR proteins, within approximately 10 min of collision with *RTS1* (*Mohebi et al., 2015*; *Nguyen et al., 2015*). Similar to BIR, Rad51 and Rad52 promote replication restart at *RTS1* and, in doing so, can inadvertently cause genomic rearrangements through the recombination of repetitive DNA elements (*Ahn et al., 2005*; *Lambert et al., 2005*; *Lambert et al., 2010*; *Mizuno et al., 2009*; *Morrow et al., 2017*; *Nguyen et al., 2015*). Also like BIR, RDR induced by *RTS1* is highly mutagenic and prone to TS (*Iraqui et al., 2012*; *Mizuno et al., 2013*; *Nguyen et al., 2015*). In this paper, we advance from our earlier study by identifying some of the key factors that influence TS during RDR, including key antirecombinogenic DNA helicases and collisions between restarted replication and tRNA transcription. We also highlight apparent similarities and differences between TS associated with BIR in *S. cerevisiae* and *RTS1*-induced RDR in *S. pombe*.

## Results

### Elevated levels of TS are detected up to 75 kb downstream of *RTS1*

In our previous study, we established an assay in *S. pombe* for studying TS associated with restart of a collapsed, yet unbroken, replication fork (*Nguyen et al., 2015*). Using this assay, we found that restarted replication is associated with high levels of TS similar to BIR in budding yeast (*Nguyen et al., 2015*). In *S. cerevisiae*, increased TS associated with BIR is only observed within the first 10 kb of template DNA that is copied, after which it drops to spontaneous levels (*Mayle et al., 2015*; *Smith et al., 2007*). In *S. pombe*, we observed high levels of TS at 12.4 kb downstream of the site of fork collapse but had not looked beyond this (*Nguyen et al., 2015*). To see if TS associated with RDR from a collapsed fork in *S. pombe* is similarly restricted to a ~ 10 kb region, as it is during BIR in *S. cerevisiae*, we used strains with a recombination reporter inserted at one of five different sites, ranging from 0.2 kb to 140 kb downstream of the *RTS1* RFB on the right arm of chromosome 3 (*Figure 1A,B*). We also positioned *RTS1* 'within' the reporter (we will refer to this as the 0 kb site) to monitor TS recombination during the initiation of RDR (*Figure 1B*). The reporter consists of a direct repeat of *ade6⁻* heteroalleles with an intervening *his3⁺* gene (*Figure 1B*). A single TS between these repeats causes a deletion of the intervening region and formation of one *ade6⁺* gene, whereas

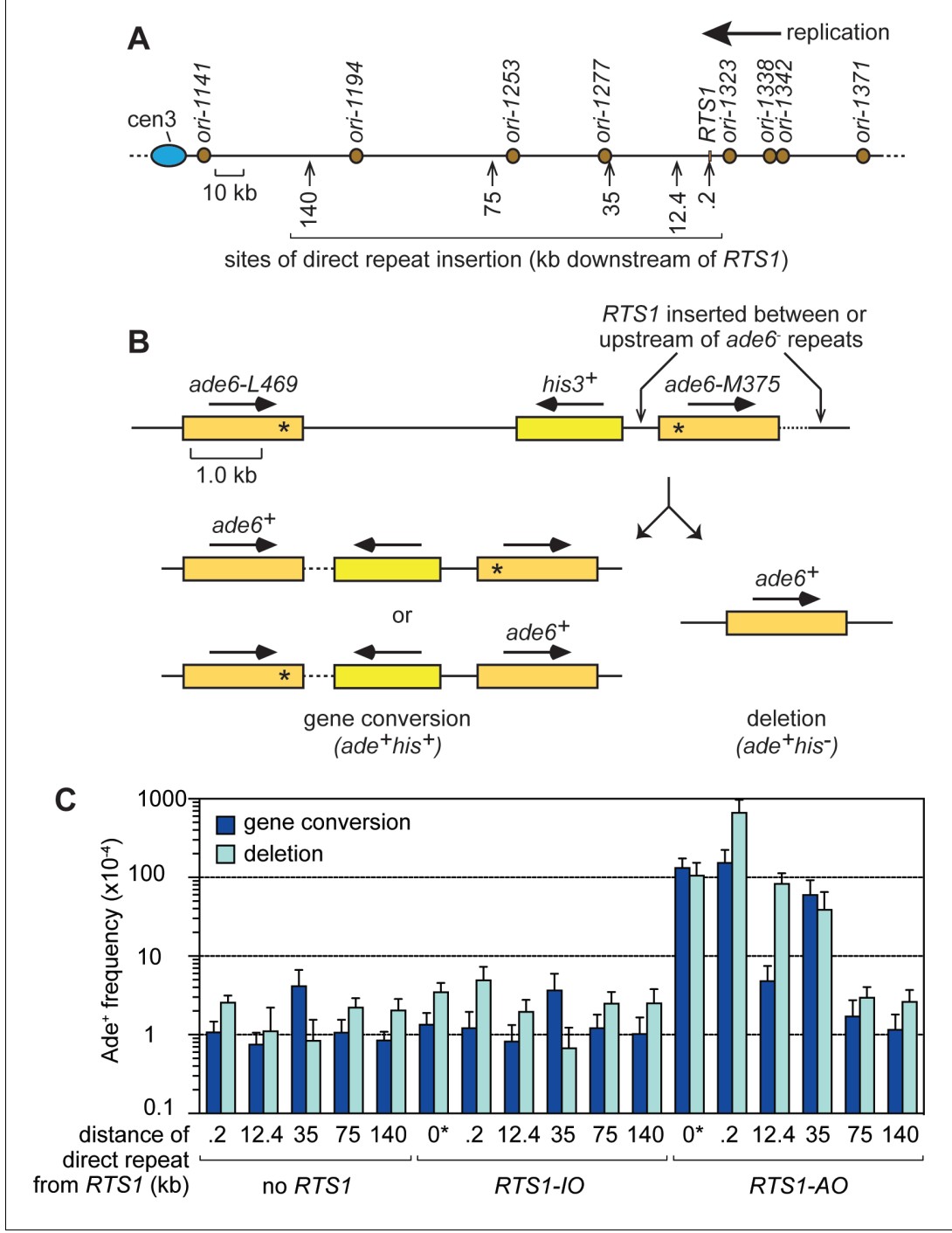

**Figure 1.** TS downstream of *RTS1*. (**A**) Map showing sites of insertion for the direct repeat recombination reporter on chromosome 3. (**B**) Schematic of the direct repeat recombination reporter showing the position of *RTS1* insertion and two types of Ade⁺ recombinant. Asterisks indicate the position of point mutations in *ade6-L469* and *ade6-M375*. (**C**) Ade⁺ recombinant frequencies for strains MCW429, MCW7229, MCW7429, MCW7430, MCW7297, MCW4712, MCW7131, MCW7257, MCW7565, MCW7614, MCW7326, MCW4713, MCW7133, MCW7259, MCW7567, MCW7616 and MCW7328. '0*' indicates that *RTS1* is positioned between the *ade6⁻* repeats. Data are mean values ± SD. Ade⁺ recombinant frequencies with statistical analysis are also shown in ***Supplementary file 1***.
DOI: https://doi.org/10.7554/eLife.41697.002

The following figure supplement is available for figure 1:

**Figure supplement 1.** Effect of *ori1253Δ* on the frequency of TS recombination 75 kb downstream of *RTS1*.
*Figure 1 continued on next page*

*Figure 1 continued*

DOI: https://doi.org/10.7554/eLife.41697.003

two TS events can convert one of the *ade6⁻* heteroalleles to *ade6⁺*. These events are distinguished by the loss/retention of the *his3⁺* gene (*Figure 1B*). *RTS1* is inserted close to a cluster of early firing replication origins (*ori-1323*, *ori-1338* and *ori-1342*), which means that it is almost always encountered first by a replication fork moving in the telomere to centromere direction (*Figure 1A*). Therefore, as *RTS1* is a polar RFB, it only blocks forks at this site in one orientation, which we refer to as the active orientation (AO) (*Nguyen et al., 2015*). The non-blocking orientation is referred to as the inactive orientation (IO) and acts as a negative control for our experiments (*Nguyen et al., 2015*). Indeed, the Ade⁺ recombinant frequencies, measured at the 35 kb, 75 kb and 140 kb sites are essentially the same with and without *RTS1-IO* (p > 0.4), and at the 0 kb, 0.2 kb and 12.4 kb sites *RTS1-IO* causes only a modest ~1.4–1.9-fold increase in deletions (p ≤ 0.001) compared to the spontaneous frequencies obtained from equivalent strains with no *RTS1* (*Figure 1C*). Compared to strains with *RTS1-IO*, those with *RTS1-AO* exhibited increases of between 32-fold to 133-fold in the frequency of Ade⁺ recombinants at sites up to 35 kb downstream of the RFB, but no increase was observed at the 75 kb and 140 kb sites (*Figure 1C*).

In our previous study, we showed that convergence with a canonical replication fork limits the distance travelled by restarted replication and, therefore, determines the chromosomal region in which TS can occur (*Nguyen et al., 2015*). To see if the absence of increased levels of TS at the 75 kb reporter was due to replication convergence occurring upstream of this site, we deleted replication origin *ori-1253*, which is responsible for more than 80% of canonical forks that normally converge with the collapsed/restarted fork (*Nguyen et al., 2015*). This deletion had no effect on the frequency of spontaneous recombination at the 75 kb reporter (p = 0.58 and 0.34 for gene conversions and deletions, respectively), whereas it caused an approximately 5-fold increase in deletions when *RTS1-AO* was present (p < 0.0001) (*Figure 1—figure supplement 1*). These data show that restarted replication can cause increased levels of TS at least as far as 75 kb downstream of a collapsed replication fork in *S. pombe*.

The increases in recombinants up to 35 kb downstream of *RTS1-AO* are comparable to the 49-fold increase observed when the RFB is placed at the 0 kb site between the two *ade6⁻* heteroalleles (*Figure 1C*). However, at the 0 kb site, the fold increase in gene conversions is higher than that of deletions (by ~3:1), whereas, at the 0.2 kb downstream site, the fold increases in gene conversions and deletions are comparable (126-fold and 135-fold, respectively) and, at the 12.4 kb, 35 kb and 75 kb sites, the fold increase in deletions (42-fold, 58-fold and 4-fold, respectively) is much higher than for gene conversions (6-fold, 16-fold and 2-fold, respectively). These data suggest that during the initial phases of restarting DNA replication there is a greater likelihood of generating a gene conversion, whereas the TS associated with restarted replication is more likely to give rise to a deletion.

## High levels of TS downstream of *RTS1-AO* depend on Pfh1

In *S. cerevisiae*, efficient BIR depends on the Pif1 DNA helicase, which aids D-loop migration (*Wilson et al., 2013*). To investigate whether Pif1 is similarly important for RDR in *S. pombe*, as it is for BIR in *S. cerevisiae*, we made use of two mutant alleles of the *S. pombe* Pif1 gene, called *pfh1-m21* and *pfh1-mt**, which encode protein that localises normally to mitochondria but has greatly reduced (*pfh1-m21*) or no detectable (*pfh1-mt**) nuclear localization (*Pinter et al., 2008*). This was important because Pfh1 has an essential mitochondrial function in *S. pombe* (*Pinter et al., 2008*). We first determined whether *pfh1-m21* and *pfh1-mt** affected the frequency of inter-repeat recombination in cells with the repeats flanking *RTS1-IO* or placed 0.2 kb or 12.4 kb downstream of it (*Figure 2A*). In the case of *pfh1-m21*, there was little or no effect on the frequency of Ade⁺ recombinants at the three sites (*Figure 2A*). The same is true for *pfh1-mt** at the 0.2 kb and 12.4 kb reporter sites (*Figure 2A*). However, when the repeats flank *RTS1-IO*, the *pfh1-mt** mutant exhibits an approximately 3-fold increase in Ade⁺ recombinants, which are mostly (87%) deletions (*Figure 2A*). This increase is more than the 2-fold increase observed in the equivalent strain with no *RTS1* (*Steinacher et al., 2012*), which suggests that Pfh1 might be needed for aiding non-recombinogenic DNA replication past *RTS1-IO*.

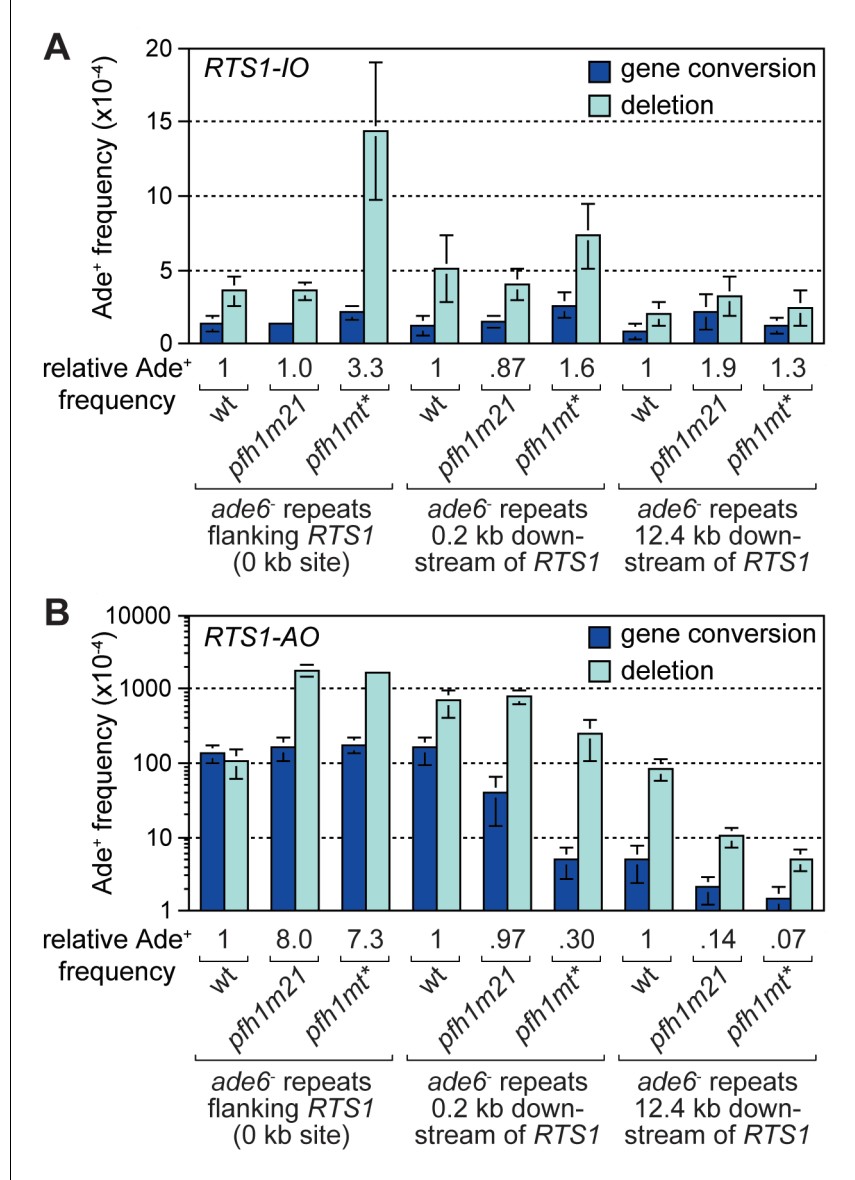

**Figure 2.** Effect of *pfh1-m21* and *pfh1-mt\** mutations on TS recombination at and downstream of *RTS1*. (**A**) Ade[+] recombinant frequencies for strains MCW4712, MCW4940, MCW4954, MCW7131, MCW7599, MCW7603, MCW7257, MCW7421 and MCW7425. (**B**) Ade[+] recombinant frequencies for strains MCW4713, MCW4942, MCW4956, MCW7133, MCW7601, MCW7605, MCW7259, MCW7422 and MCW7426. Data are mean values ± SD. Ade[+] recombinant frequencies with statistical analysis are also shown in **Supplementary file 1**.

DOI: https://doi.org/10.7554/eLife.41697.004

The following figure supplement is available for figure 2:

**Figure supplement 1.** Relative Ade[+] recombinant frequencies for the *pfh1-m21* and *pfh1-mt\** strains shown in **Figure 2B**.

DOI: https://doi.org/10.7554/eLife.41697.005

We next determined the recombination frequency in strains with *RTS1-AO* (**Figure 2B**). Consistent with previous data (**Steinacher et al., 2012**), we observed a greater than 7-fold increase in Ade[+] recombinants in a *pfh1-mt\** mutant, compared to wild-type, when *RTS1-AO* is flanked by the repeats at the 0 kb site (**Figure 2B**, **Figure 2—figure supplement 1**). A similar increase was seen in a *pfh1-m21* mutant and, in both cases, essentially all of the additional recombinants were deletions (**Figure 2B**, **Figure 2—figure supplement 1**). In contrast, the frequency of recombinants formed at

the 0.2 kb and 12.4 kb reporter sites (by TS associated with replication restart) was reduced in both mutants (*Figure 2B*, *Figure 2—figure supplement 1*). Importantly, there were bigger fold reductions at the 12.4 kb reporter than at the 0.2 kb reporter, and with the *pfh1-mt** mutant compared to the *pfh1-m21* mutant (*Figure 2B*, *Figure 2—figure supplement 1*). These data show that there is a doseage-sensitive requirement for Pfh1 in promoting TS recombination downstream of *RTS1-AO*. This requirement might reflect a direct role in TS or a role in promoting replication restart.

## Pfh1 promotes replication restart at *RTS1-AO*

To determine whether Pfh1 is needed for RDR at *RTS1*, we analysed DNA replication intermediates in a ~ 3.7 kb Sac1 restriction fragment, that encompasses the barrier, by native two-dimensional gel eletrophoresis (2DGE) (*Figure 3A,B*). As seen previously, replication forks accumulated at *RTS1-AO* in both wild-type and *pfh1-mt** mutant (*Figure 3B*) (*Steinacher et al., 2012*). However, whilst in the wild-type blocked forks were resolved through a mixture of restart (indicated by the presence of large Y-shaped molecules) and fork convergence at the RFB (indicated by the presence of double Y-shaped molecules), in the *pfh1-mt** mutant they were resolved mainly by fork convergence (indicated by a reduction in large Y-shaped molecules and increase in double Y-shaped molecules) (*Figure 3B,C*). These data indicate that Pfh1 promotes efficient replication restart following fork collapse at *RTS1*.

Intriguingly, we also observed the accumulation of a novel DNA signal running approximately parallel to the ascending arc of small Y-shaped molecules in the *pfh1-mt** mutant (*Figure 3B*, labelled 'e' in the left hand panel). A similar signal was recently reported to accumulate in samples from *rad51Δ* and *rad52Δ* mutants, and was shown to depend on the 5' to 3' exonuclease Exo1 (*Ait Saada et al., 2017*). Based on these findings, it was proposed that Rad51 and Rad52 protect the collapsed replication fork from excessive Exo1 activity, which could generate extensively resected regressed replication forks (*Ait Saada et al., 2017*; *Teixeira-Silva et al., 2017*). To see if the novel DNA signal from *pfh1-mt** depended on Exo1, we analysed replication intermediates from a *pfh1-mt** *exo1Δ* double mutant (*Figure 3B*). This analysis showed that the novel signal was indeed Exo1 dependent. Exactly how Pfh1 prevents accumulation of resected replication forks is unclear. It may simply be a byproduct of its role in promoting efficient replication restart. Alternatively, Pfh1 may play a direct role in protecting the collapsed fork. Regardless of the mechanism, the similarly low levels of large Y-shaped DNA molecules, from both the *pfh1-mt** single mutant and *pfh1-mt** *exo1Δ* double mutant, indicate that Pfh1 does not simply counter Exo1 activity to promote efficient RDR.

## Pfh1 limits TS downstream of *RTS1-AO*

The discovery that Pfh1 is needed to promote efficient RDR, explains why TS downstream of *RTS1-AO* is reduced in *pfh1-m21* and *pfh1-mt** mutants. This finding also opened up the possibility that Pfh1's function in promoting efficient RDR might mask a role for it in limiting TS downstream of *RTS1-AO*. To investigate this possibility, we tested the effect of deleting *ori-1253* on the frequency of TS at 0.2 and 12.4 kb reporter sites in the two *pfh1* mutants, reasoning that allowing more time for RDR might partially offset the need for Pfh1. In the control strains, with *RTS1-IO*, a modest increase in the frequency of Ade[+] recombinants was observed at the 0.2 kb reporter (but not 12.4 kb reporter) in both *pfh1* mutants, which might stem from fork collapse at *RTS1-IO* due to loss of Pfh1 'sweepase' activity (see Discussion) (*Figure 4*). In the strains with *RTS1-AO*, the frequency of Ade[+] recombinants increased at both reporter sites when *ori-1253* was deleted and, whilst the frequency of gene conversions remained less in the *pfh1* mutants than in wild-type, the overall number of recombinants in the *pfh1-m21* mutant was more than 2-fold higher than wild-type at the 0.2 kb reporter due to a greater increase in deletions (*Figure 4*, *Figure 4—figure supplement 1*). These data indicate that Pfh1, in addition to promoting RDR, also limits TS recombination.

## A tRNA gene increases TS recombination downstream of *RTS1-AO*

Collisions between replication forks and RNA Polymerase III (Pol III) transcription complexes at tRNA genes are known to stall fork progression (*Deshpande and Newlon, 1996*; *Pryce et al., 2009*). However, unlike at *RTS1*, fork stalling at tRNA genes generally only elicits detectable changes in recombination when in conjunction with other genetic elements that perturb DNA replication, or when there are defects in the replication machinery (reviewed in *McFarlane and Whitehall, 2009*).

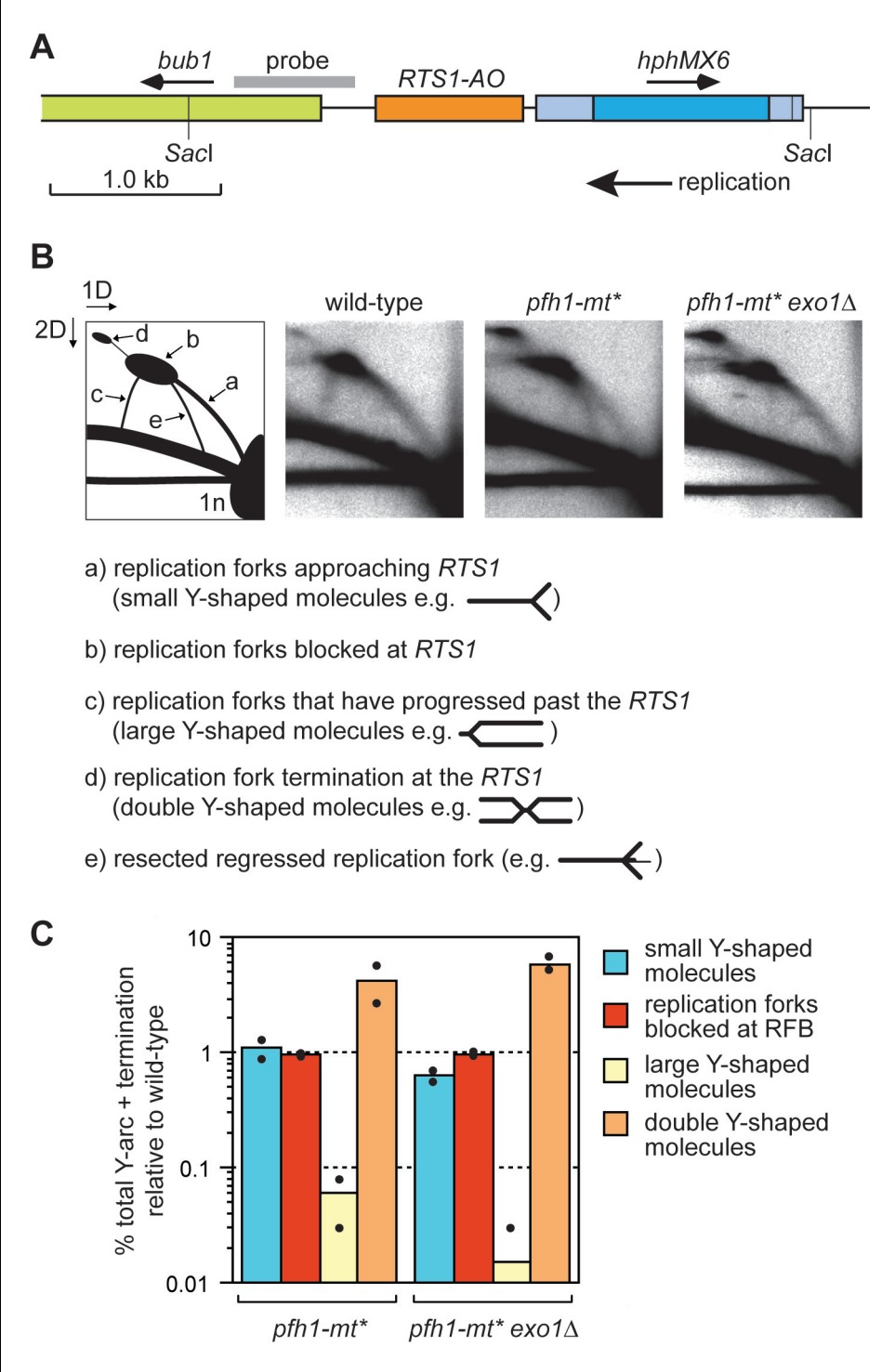

**Figure 3.** Pfh1 is needed for replication past *RTS1-AO*. (**A**) Schematic showing location of *RTS1-AO* and *hphMX6* adjacent to *bub1* on chromosome 3. The position of the probe used for the 2DGE analysis in B) is also shown. (**B**) 2DGE of replication intermediates in the SacI fragment shown in **A**). The DNA was extracted from strains MCW7223, MCW8587 and MCW8605. (**C**) Quantification of 2DGE. Values are relative to wild-type and are based on two independent experiments with each value represented by a dot around the mean.
DOI: https://doi.org/10.7554/eLife.41697.006

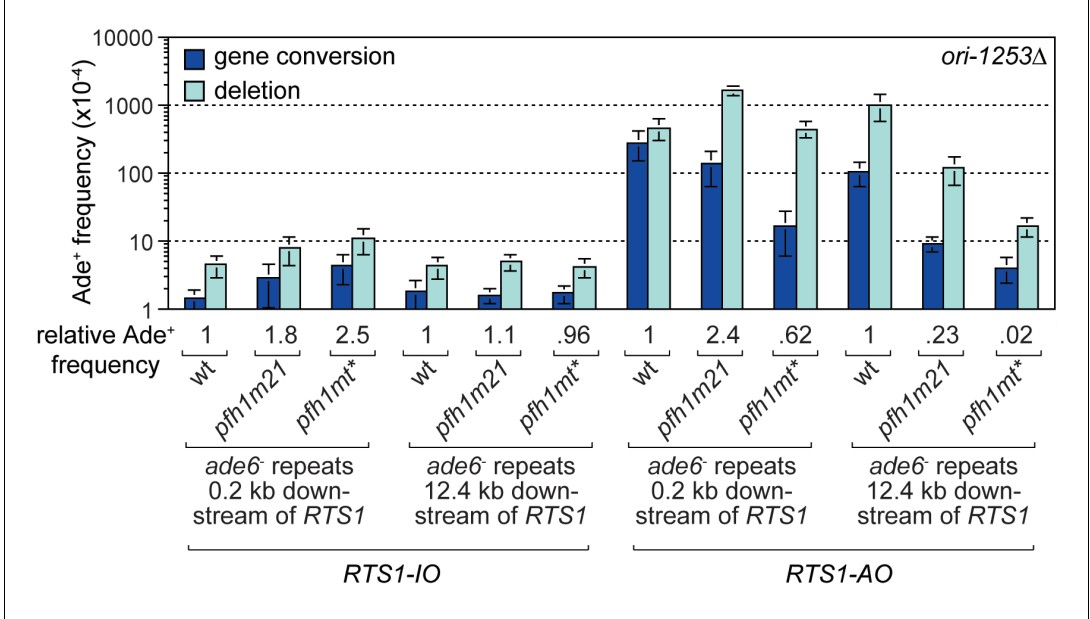

**Figure 4.** Effect of *ori1253Δ* on the frequency of TS recombination downstream of *RTS1* in wild-type and *pfh1* mutants. The strains are MCW7414, MCW7598, MCW7602, MCW7293, MCW7423, MCW7427, MCW7416, MCW7600, MCW7604, MCW7295, MCW7424, MCW7428. Data are mean values ± SD. Ade$^+$ recombinant frequencies with statistical analysis are also shown in **Supplementary file 1**.

DOI: https://doi.org/10.7554/eLife.41697.007

The following figure supplement is available for figure 4:

**Figure supplement 1.** Relative Ade$^+$ recombinant frequencies for the *pfh1-m21* and *pfh1-mt\** strains shown in **Figure 4**.

DOI: https://doi.org/10.7554/eLife.41697.008

For example, work from our lab previously showed that loss of Pfh1 causes a ~ 27-fold increase in direct repeat recombination when DNA replication is stalled through head-on collision with transcription at a tRNA gene inserted between the DNA repeats (**Steinacher et al., 2012**). Whether this is also true for collisions between RDR and Pol III complexes has previously not been examined. To investigate the potential for TS to be triggered by collisions between restarted replication and Pol III complexes, we inserted a *tRNA$^{GLU08}$* gene between the *ade6$^-$* heteroalleles at the 12.4 kb reporter site downstream of *RTS1-AO* (**Figure 5A**). As the orientation of collision between replication and transcription complex can influence stalling and associated genome instability (**Osmundson et al., 2017**; **Steinacher et al., 2012**; **Tran et al., 2017**), we constructed yeast strains with *tRNA$^{GLU08}$* inserted so that collision between restarted replication and transcription complex would either be co-directional (*tRNA$^{GLU08}$CD*) or head-on (*tRNA$^{GLU08}$HO*) (**Figure 5A**). In the absence of *RTS1-AO*, there was little difference in the recombination frequency between strains with and without *tRNA$^{GLU08}$CD*, whereas with *tRNA$^{GLU08}$HO* there was a modest 1.5-fold increase in Ade$^+$ recombinants (**Figure 5B**). A similar pattern was seen in the strains with *RTS1-AO*, although in this case the fold increase in Ade$^+$ recombinants with *tRNA$^{GLU08}$HO* was higher (2.3-fold) (**Figure 5C**). These data indicate that head-on collision between a restarted replication complex and a Pol III complex can trigger TS.

We next investigated whether loss of Pfh1 would further exacerbate TS induced by *tRNA$^{GLU08}$HO*. In the absence of *RTS1*, both *pfh1-m21* and *pfh1-mt\** mutants caused a marked increase in Ade$^+$ recombinants (17.3-fold and 31.6-fold, respectively) consistent with previous findings (**Steinacher et al., 2012**) (**Figure 5D**). However, in the presence of *RTS1-AO*, the fold change in TS in *pfh1-m21* and *pfh1-mt\** mutants compared to wild-type was essentially the same with and without *tRNA$^{GLU08}$HO* (compare data in **Figure 2B** and **Figure 5D**). These data suggest that Pfh1 is not needed to suppress TS induced by head-on collision between a restarted replication complex and a Pol III complex.

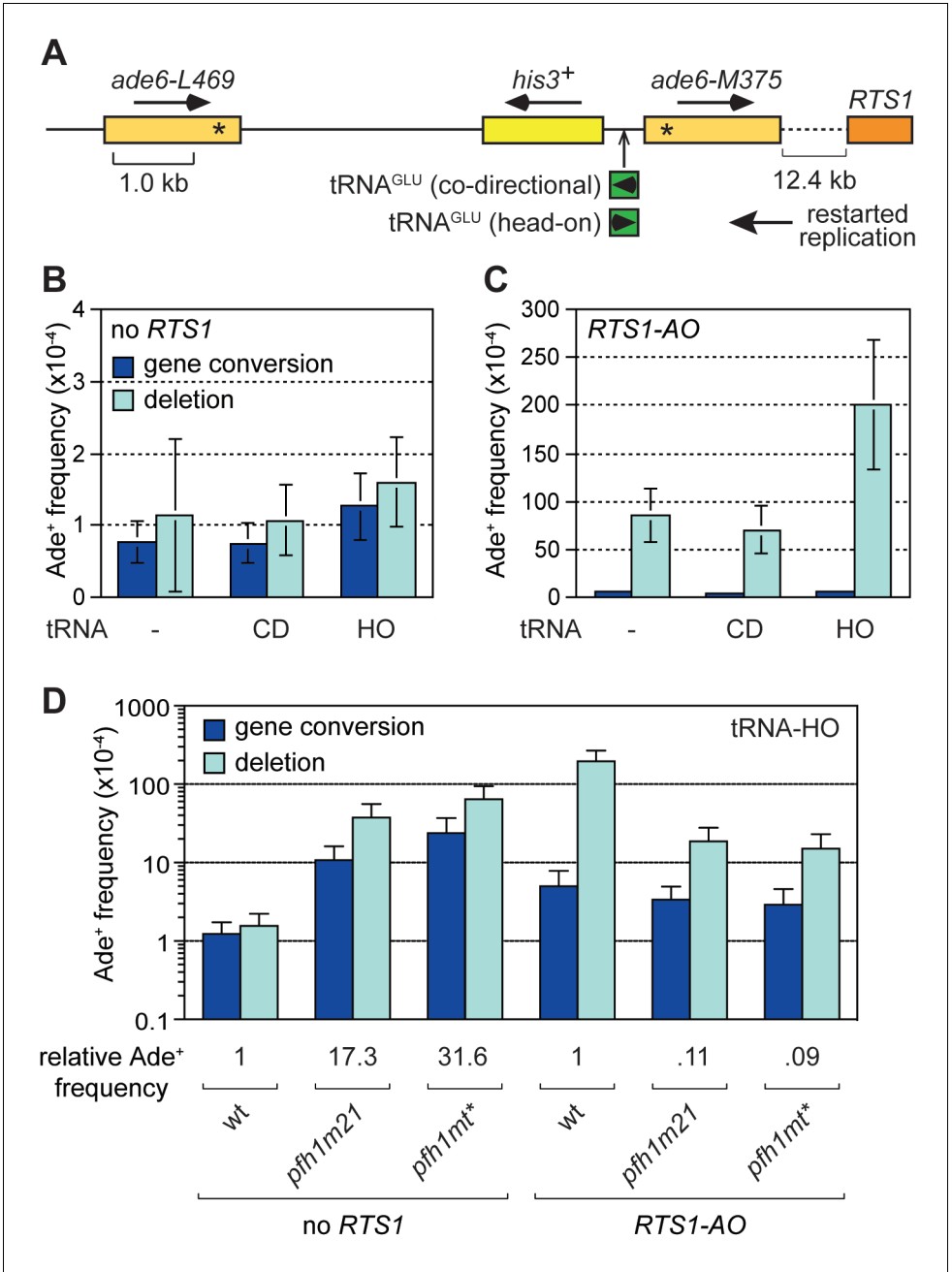

**Figure 5.** Effect of *tRNA^GLU08^CD* and *tRNA^GLU08^HO* on TS downstream of *RTS1*. (**A**) Schematic showing the position of *tRNA^GLU08^CD/HO* within the direct repeat recombination reporter 12.4 kb downstream of *RTS1*. (**B**) Ade^+^ recombinant frequencies for strains MCW7229, MCW7434 and MCW7433. (**C**) Ade^+^ recombinant frequencies for strains MCW7259, MCW7521 and MCW7517. (**D**) Ade^+^ recombinant frequencies for strains MCW7433, MCW9381, MCW9383, MCW7517, MCW9360 and MCW9361. Data are mean values ± SD. Ade^+^ recombinant frequencies with statistical analysis are also shown in *Supplementary file 1*.
DOI: https://doi.org/10.7554/eLife.41697.009

## Rqh1 limits TS downstream of *RTS1-AO*

The RecQ family DNA helicase Rqh1 has been shown to limit ectopic recombination at the *RTS1* RFB (*Ahn et al., 2005*; *Lambert et al., 2010*; *Lorenz et al., 2009*; *Pietrobon et al., 2014*). To investigate whether it also affects TS associated with restarted DNA replication, we compared the frequency of Ade^+^ recombinants in a wild-type and *rqh1Δ* mutant at the 12.4 kb reporter site (*Figure 6C*). This

analysis revealed a 2.9-fold increase in recombinants in the mutant, indicating that Rqh1 plays a role in limiting TS. However, whereas the fold increase in recombinants at 12.4 kb is similar to that for spontaneous recombination at the 0 kb reporter site (*Figure 6A,C*), it is significantly less than the 10-fold increase seen at the 0 kb site when *RTS1-AO* is flanked by the ade6⁻ repeats (*Figure 6B*, *Figure 6—figure supplement 1*). Rqh1 is not required for replication restart at *RTS1* (*Lambert et al., 2010*) and, therefore, the lower fold increase in recombinants at the 12.4 kb reporter site, compared to the 0 kb site, suggests that Rqh1 plays a more important role in limiting ectopic recombination during replication restart than in controlling TS associated with restarted replication.

## CAF1 promotes TS

It has been proposed that Rqh1 limits ectopic recombination at the *RTS1* RFB by unwinding D-loops formed by Rad51, and that this activity is counteracted by chromatin assembly factor 1 (CAF1) (*Pietrobon et al., 2014*). Our analysis of null mutants of two of the three CAF1 subunits (Pcf2 and Pcf3) reveals little or no change in spontaneous recombination at the 0 kb reporter but an approximately 3-fold reduction in gene conversions and ~30% reduction in deletions when *RTS1-AO* is present (*Figure 6A,B*, *Figure 6—figure supplement 1*). We also observed a 2-fold reduction in gene conversions and a 4-fold reduction in deletions in both *pcf2Δ* and *pcf3Δ* mutants downstream of *RTS1-AO* at the 12.4 kb reporter (*Figure 6C*, *Figure 6—figure supplement 1*). These data confirm that CAF1 promotes ectopic recombination associated with restarting replication at the *RTS1* RFB. Moreover, considering that CAF1 is not required for replication restart (*Pietrobon et al., 2014*), our data show that it also promotes TS associated with restarted replication.

## Srs2 and Fbh1 suppress TS associated with restarted replication

The DNA helicases Srs2 and Fbh1, which are members of the UvrD family of proteins, are known regulators of Rad51-mediated recombination, especially at sites of replication fork stalling and collapse (*Chiolo et al., 2007*; *Fugger et al., 2009*; *Lorenz et al., 2009*; *Marini and Krejci, 2010*; *Morishita et al., 2005*; *Osman et al., 2005*; *Simandlova et al., 2013*; *Tsutsui et al., 2014*). Although a previous study showed that both proteins limit ectopic recombination at the *RTS1* RFB (*Lorenz et al., 2009*), it was not known whether they have a similar influence on TS associated with restarted replication. To investigate this, we compared the frequency of inter-repeat recombination at and downstream of *RTS1-AO* in both *srs2Δ* and *fbh1Δ* mutants (*Figure 6B,C*). In the case of *srs2Δ*, there was a similar 6-fold increase in Ade⁺ recombinants at both 0 kb and 12.4 kb reporters, which was also similar to the fold increase in spontaneous recombination at the 0 kb reporter containing *RTS1-IO* (*Figure 6A,B,C*). Unlike a *srs2Δ* mutant, *fbh1Δ* does not exhibit an increase in spontaneous recombination at the 0 kb reporter (*Figure 6A*) (*Lorenz et al., 2009*). However, when *RTS1-AO* is present, *fbh1Δ* causes an even greater increase in Ade⁺ recombinants at both 0 kb and 12.4 kb reporters than *srs2Δ* (*Figure 6B,C*). Altogether these data show that both Srs2 and Fbh1 strongly inhibit TS recombination associated with restarted replication. It has been reported that Srs2 is needed to promote efficient replication restart at *RTS1* (*Inagawa et al., 2009*; *Lambert et al., 2010*) and, therefore, the increase in TS we observe at the 12.4 kb reporter in a *srs2Δ* mutant may be an underestimate of Srs2's importance in suppressing TS.

## Deleting *fml1* or *mus81* has little or no effect on the frequency of TS recombination downstream of *RTS1-AO*

In *S. cerevisiae*, the FANCM-related DNA helicase Mph1 and structure-specific nuclease Mus81 have both been shown to affect levels of TS associated with BIR, with the former promoting TS and the latter limiting it (*Mayle et al., 2015*; *Stafa et al., 2014*). To see if *S. pombe* Mph1 (termed Fml1) and Mus81 similarly affect TS associated with replication restart at *RTS1*, we compared the frequency of inter-repeat recombination at and downstream of *RTS1-AO* in both *fml1Δ* and *mus81Δ* mutants (*Figure 6B,C*). As reported previously, loss of *fml1* results in an almost 10-fold reduction in gene conversions at the 0 kb reporter whilst the frequency of deletions remains unchanged (*Figure 6B*, *Figure 6—figure supplement 1*) (*Sun et al., 2008*). Spontaneous recombination at the 0 kb reporter is similarly affected by *fml1Δ*, albeit the fold reduction in gene conversions (~2.6-fold) is not as great (*Figure 6A*). In contrast, there is little change in the frequency of Ade⁺ recombinants at the 12.4 kb reporter downstream of *RTS1-AO* (*Figure 6C*, *Figure 6—figure supplement 1*).

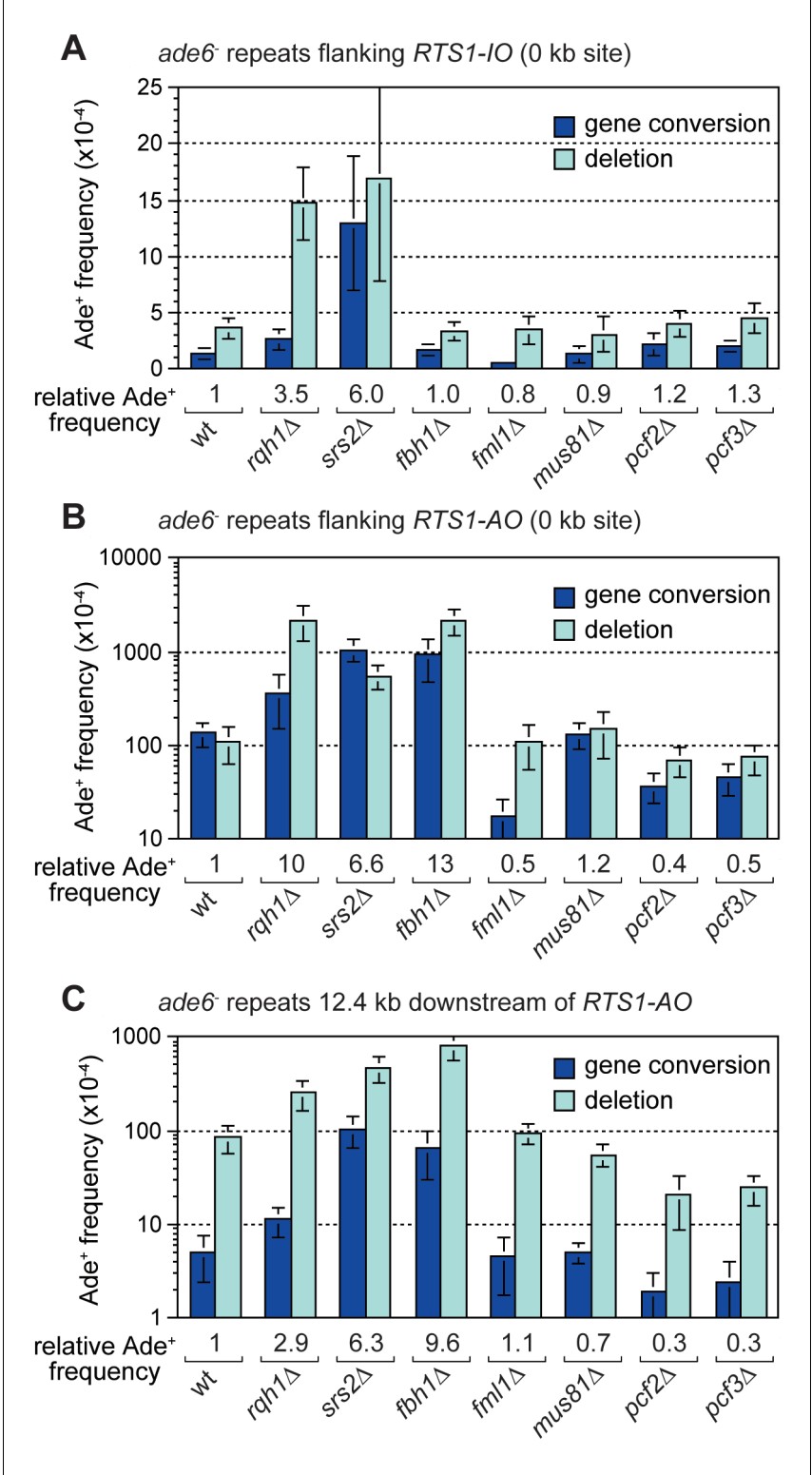

**Figure 6.** Effect of *rqh1Δ*, *srs2Δ*, *fbh1Δ*, *fml1Δ*, *mus81Δ*, *pcf1Δ* and *pcf2Δ* on TS recombination at and downstream of *RTS1*. (**A**) Ade$^+$ recombinant frequencies for strains MCW4712, MCW1443, FO1748, FO1814, MCW3059, MCW1451, MCW6972 and MCW7147. (**B**) Ade$^+$ recombinant frequencies for strains MCW4713, MCW1447, FO1750, FO1816, MCW3061, MCW1452, MCW7213 and MCW7149. (**C**) Ade$^+$ recombinant frequencies for strains MCW7259, MCW8201, MCW8200, MCW8227, MCW8193, MCW8195, MCW8359 and MCW8360. Data are mean values ± SD. Ade$^+$ recombinant frequencies with statistical analysis are also shown in *Supplementary file 1*.

*Figure 6 continued on next page*

*Figure 6 continued*

DOI: https://doi.org/10.7554/eLife.41697.010

The following figure supplements are available for figure 6:

**Figure supplement 1.** Relative Ade⁺ recombinant frequencies for the *RTS1-AO* containing strains shown in *Figure 5*.

DOI: https://doi.org/10.7554/eLife.41697.011

**Figure supplement 2.** Fml1 and Mus81 are not needed for replication past *RTS1-AO*.

DOI: https://doi.org/10.7554/eLife.41697.012

Deletion of *mus81* has no effect on the frequency of gene conversions at either 0 or 12.4 kb reporters, but does cause a slight increase in deletions (~1.4-fold) at the 0 kb reporter and a decrease in deletions (~1.5-fold) at the 12.4 kb reporter (*Figure 6BC*, *Figure 6—figure supplement 1*). The absence of any major change in TS frequency at the 12.4 kb reporter, in both the *fml1Δ* and *mus81Δ* mutant, could be due to a change in restart efficiency offsetting a potential decrease or increase in TS. However, 2DGE of replication intermediates at and around *RTS1-AO* reveals wild-type levels of large Y-shaped and double Y-shaped DNA molecules, suggesting that restart is unaffected by loss of either Fml1 or Mus81 (*Figure 6—figure supplement 2*).

## Discussion

The factors affecting TS associated with recombination-mediated DNA replication have hitherto been studied mainly in the context of BIR in *S. cerevisiae* and outside of S-phase. In our study we have measured TS associated with RDR from a collapsed, yet unbroken, replication fork in *S. pombe*. With this system, we have identified similarities with BIR in *S. cerevisiae*, as well as some potential differences. To place our findings into a conceptual framework, we will first discuss a hypothetical model for TS, which is elaborated from a previous model (*Figure 7*) (*Nguyen et al., 2015*). We envisage that the collapsed replication fork regresses to form a classic 'chicken foot' structure with an exposed ssDNA tail formed by the nucleolytic activity of the Mre11-Rad50-Nbs1-Ctp1 complex and Exo1 (*Figure 7*, step 1) (*Teixeira-Silva et al., 2017*). Rad51 loads onto the ssDNA tail to form a nucleoprotein filament that first locates a homologous duplex and then invades it to form a D-loop (*Figure 7*, step 2). Polδ extends the 3' end of the invading strand (*Miyabe et al., 2015*) and, as it does, the D-loop moves in tandem (*Figure 7*, step 3) (*Saini et al., 2013*). Although not shown in *Figure 7*, lagging strand DNA synthesis may be established on the nascent leading strand giving rise to conservative replication, which could then be directed into semi-conservative replication by branch migration of the regressed replication fork behind the D-loop (*Nguyen et al., 2015*). Such a model would be compatible with the finding that RDR, triggered by fork collapse at *RTS1*, is semi-conservative (*Miyabe et al., 2015*), and contrasts with BIR, which is conservative (*Donnianni and Symington, 2013*; *Saini et al., 2013*). As DNA replication progresses, the associated migrating D-loop is likely to be susceptible to dissociation, either through the action of DNA helicases and/or by branch migration of the regressed replication fork 'catching up' with it (*Nguyen et al., 2015*). Dissociation of the D-loop in a region of repetitive DNA would lead to TS if the ejected DNA strand re-invades the 'wrong' repeat (*Figure 7*, steps 4 and 5; note that the DNA repeats, in yellow, are directly orientated as they are in our genetic assay). RDR is terminated upon encounter with the oncoming replication fork (*Figure 7*, steps 6 and 7) and, during this process, the invading DNA strand may either be ligated to the nascent lagging strand of the oncoming fork (*Figure 7*, step 7a) or be unwound (*Figure 7*, step 7b). If unwound, a second TS may be triggered (not shown) or the regressed replication fork may simply re-set by branch migration and merge with the oncoming fork (*Figure 7*, steps 8b and 9b). Note that these events can give rise to a gene conversion in our genetic assay (*Figure 1B*). Ligation of the invading DNA strand to the nascent lagging strand of the oncoming fork would prevent further TS and consolidate the formation of a deletion, with branched DNA structures being resolved by nucleases (not shown) (*Lorenz et al., 2009*) or by the completion of DNA replication (*Figure 7*, steps 8a and 9a). In the latter case, the region being deleted would form a heterologous loop of ssDNA (*Figure 7*, step 9a), which could be removed by a large DNA loop repair pathway (*Clikeman et al., 2001*; *Sommer et al., 2008*).

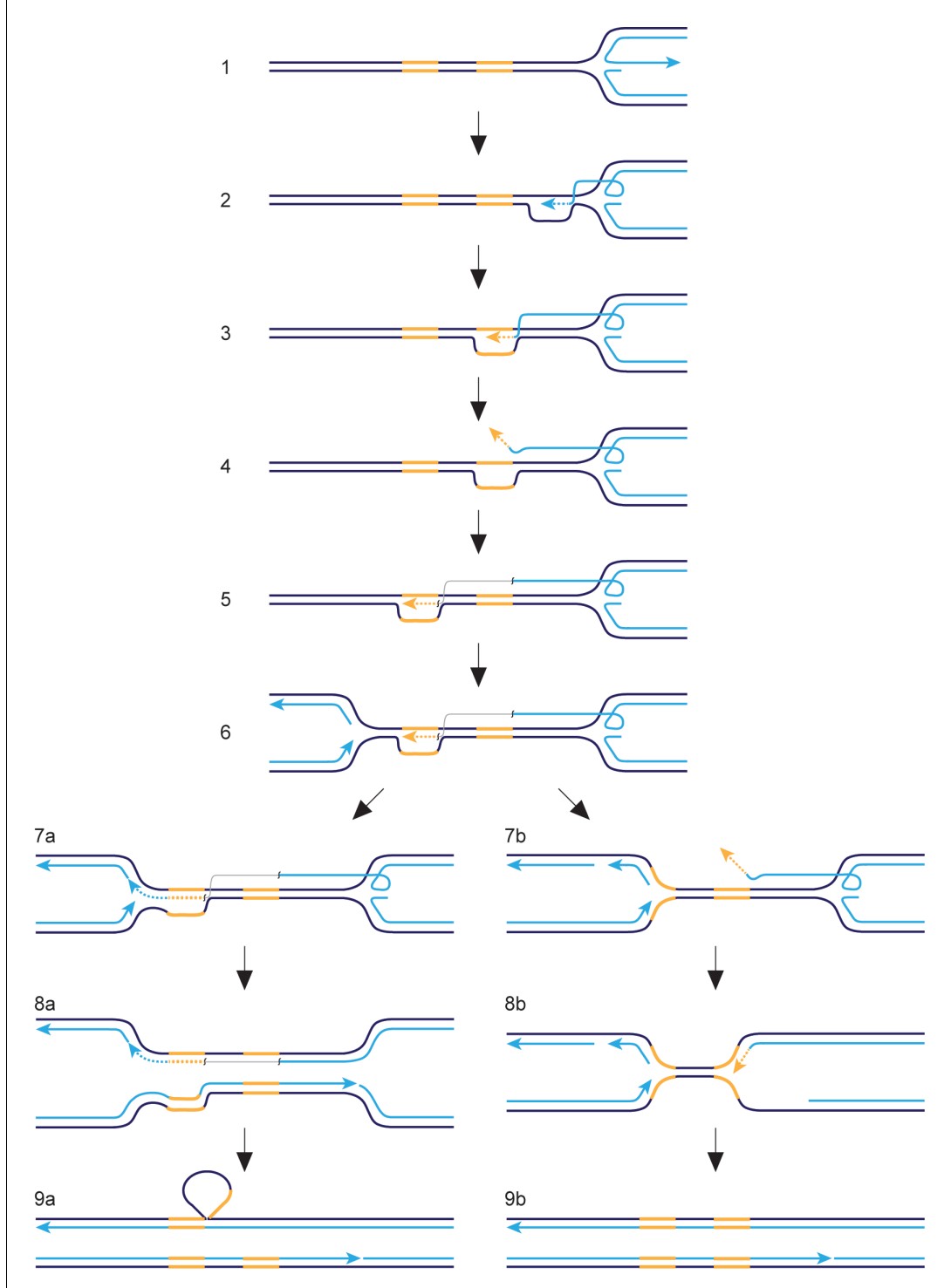

**Figure 7.** Model for TS between direct repeats. Parental DNA strands are dark blue, nascent DNA strands are light blue and DNA repeats are yellow. The grey line in steps 5, 6, 7a and 8a indicates that the light blue and yellow lines, which it connects, are continuous. See main text for a step-by-step description of the model.

DOI: https://doi.org/10.7554/eLife.41697.013

TS associated with BIR in *S. cerevisiae* is thought to reflect an inherent instability of the migrating D-loop, which has evolved to encourage DSB repair by synthesis-dependent strand annealing (*Smith et al., 2007*). The observation that TS in *S. cerevisae* is constrained to the first 10 kb of template DNA that is copied, suggests that the D-loop eventually matures into a more stable structure (*Mayle et al., 2015*; *Smith et al., 2007*). This maturation process may involve cleavage of the D-loop by structure-specific nucleases such as Mus81 (*Mayle et al., 2015*; *Osman and Whitby, 2007*). Our finding that TS downstream of *RTS1-AO* is generally biased in favour of deletions, whereas at the RFB gene conversions predominate, suggests that the frequency of double TS events declines soon after replication has restarted. Interestingly, Carr and colleagues reported that the propensity for restarted replication to perform a U-turn at small inverted repeats positioned downstream of *RTS1* declined 4-fold when the distance between RFB and repeats was increased from 0 to 2 kb, and then appeared to plateu (*Mizuno et al., 2013*). These observations suggest that the putative migrating D-loop, which is formed following replication fork collapse at *RTS1-AO*, undergoes partial stabilization shortly after it is formed. However, unlike with BIR in *S. cerevisiae*, a degree of D-loop instability must be retained because we can detect TS over distances of at least 75 kb downstream of the site of fork collapse. Our finding, that *tRNA^{GLU08}*HO increases TS downstream of *RTS1-AO*, suggests that head-on collisions with transcription complexes might exacerbate D-loop instability and be a key trigger in promoting TS.

The apparent disparity in D-loop maturation, between BIR in *S. cerevisiae* and RDR in *S. pombe*, may reflect differences in D-loop processing. In *S. cerevisiae*, Mus81 constrains TS during BIR presumably by resolving the D-loop into a more stable fork structure (*Mayle et al., 2015*). However, in *S. pombe*, we found no evidence that Mus81 is required to constrain TS during RDR. In fact the only constraint, over the region in which TS can occur in *S. pombe*, seems to be the point at which the restarted replication converges with the oncoming replication fork - a property which is shared with BIR in *S. cerevisiae* (*Mayle et al., 2015*; *Nguyen et al., 2015*).

TS associated with restarted replication is thought to be responsible for generating many of the genome rearrangements that are associated with cancer and other disease states (*Carvalho and Lupski, 2016*; *Hastings et al., 2009*). However, some complex genome rearrangements can span distances of hundreds of kilobases and, therefore, could not be formed from a single restarted replication event if TS was limited to a 10 kb region downstream of the site of fork collapse (*Hastings et al., 2009*). Our finding, that TS events occur over a much greater distance in *S. pombe* than they do for BIR in *S. cerevisiae*, shows that TS, associated with the repair of a single collapsed replication fork, may be capable of generating some of the long-range and complex genome rearrangements seen in humans.

Similar to BIR in *S. cerevisiae*, the Pif1 family DNA helicase Pfh1 promotes RDR, which explains why there is a general reduction in the levels of TS downstream of *RTS1-AO* in *pfh1-m21* and *-mt\** mutants. In *S. cerevisiae*, Pif1 aids DNA synthesis during BIR by driving D-loop migration (*Wilson et al., 2013*), and Pfh1 might do likewise during RDR in *S. pombe*. Moreover, our observation, that resected regressed replication forks accumulate at *RTS1-AO* in a *pfh1-mt\** mutant, suggests that Pfh1 is a candidate for promoting the putative branch migration and/or re-setting of the regressed replication fork behind the migrating D-loop (see Discussion above). Pfh1 also has the ability to drive DNA replication by dislodging protein-DNA barriers via its so-called sweepase activity (*Sabouri et al., 2012*; *Steinacher et al., 2012*). Efficient RDR past *RTS1* presumably requires removal of the barrier protein Rtf1, which may depend on Pfh1's sweepase activity.

Despite Pfh1 being needed for efficient RDR, we were able to detect an increase in TS in a *pfh1-m21* mutant if we provided more time for replication restart to occur, which shows that Pfh1, in addition to promoting RDR, also plays a role in suppressing TS. This finding contrasts with BIR in *S. cerevisiae* where Pif1 has no affect on TS (*Stafa et al., 2014*). Unlike *S. pombe* and humans, which have only one Pif1 family helicase, *S. cerevisiae* has a second Pif1 helicase called Rrm3, which, whilst seemingly not required for BIR, is important for the repair of replication-associated DSBs as well as replication past stable protein-DNA complexes (*Ivessa et al., 2003*; *Muñoz-Galván et al., 2017*; *Wilson et al., 2013*). It is possible that Pif1 and Rrm3 act redundantly in the suppression of TS in *S. cerevisiae*. Alternatively, the role of Pif1 helicases in suppressing TS may only be evident when RDR acts during S-phase alongside ongoing canonical DNA replication. Specifically, they may play an important role in directing the termination of DNA replication where a canonical fork encounters a D-loop. We speculate that Pfh1 drives D-loop dissociation during termination through its action as

an accessory helicase at the oncoming DNA replication fork, where it likely translocates on the lagging template strand in a 5' – 3' direction (i.e. towards the D-loop and on the same DNA strand that the invading strand is annealed to) (*Sabouri, 2017*) (*Figure 7*, step 7b). Without this activity there would be a greater tendency for the 3' end of the invading strand at the D-loop to ligate to the lagging strand of the oncoming fork, which would specifically favour deletion formation in our genetic assay (*Figure 7*, steps 7a – 9a). Indeed, in the instances where we can detect an increase in Ade[+] recombinants when *pfh1* is mutated, it is specifically deletions that increase, with the frequency of gene conversions either decreasing or remaining unchanged (*Figure 2*, *Figure 2—figure supplement 1*, *Figure 4*, *Figure 4—figure supplement 1*).

In addition to Pfh1, we have identified a further three conserved DNA helicases that suppress TS downstream of *RTS1-AO*, namely Rqh1, Srs2 and Fbh1. These helicases, and their homologues in other organisms, are well known as major players in directing the outcome of homologous recombination reactions (*Doe and Whitby, 2004*; *Fugger et al., 2009*; *Kohzaki et al., 2007*; *Larsen and Hickson, 2013*; *Lorenz et al., 2009*; *Morishita et al., 2005*; *Niu and Klein, 2017*; *Osman et al., 2005*; *Simandlova et al., 2013*; *Tsutsui et al., 2014*). However, their role in suppressing TS downstream of a collapsed replication fork has not previously been demonstrated, though it has been suggested that the homologue of Rqh1 in *S. cerevisiae* (Sgs1) might limit translocations between highly diverged genes by suppressing TS during BIR (*Schmidt et al., 2006*). Based on the known activities of their homologues, Rqh1, Srs2 and Fbh1 have the potential to influence TS in various ways. For example, Sgs1 and its human homologue BLM, as well as *S. cerevisiae* Srs2, are capable of dissociating D-loops in vitro (*Bachrati et al., 2006*; *Bugreev et al., 2007*; *Dupaigne et al., 2008*; *Fasching et al., 2015*; *Liu et al., 2017*; *van Brabant et al., 2000*). The dissociation of a nascent D-loop can abort a TS event before a genetic change is made (e.g. a deletion or gene conversion in our genetic assay) and, therefore, the loss of this activity might account for the increase in TS we observe in both *rqh1Δ* and *srs2Δ* mutants (*Figure 5*). Indeed, Lambert and colleagues showed that Rqh1 was responsible for the disappearance of recombination intermediates at *RTS1* in cells lacking CAF1, and proposed that this was due to its ability to drive D-loop dissociation (*Pietrobon et al., 2014*).

Another way in which Rqh1 may affect TS is by catalyzing the branch migration of the regressed replication fork behind the migrating D-loop. The regressed fork forms a four-way/Holliday junction, which is a structure that several RecQ–type helicases have been shown to unwind/branch migrate in vitro (*Cejka and Kowalczykowski, 2010*; *Karow et al., 2000*; *LeRoy et al., 2005*; *Machwe et al., 2011*). As discussed above, branch migration of the four-way junction into the 'back' of the D-loop would cause the D-loop to unwind. This might happen more frequently during an attempted TS event if the four-way junction continues to migrate whilst the D-loop is being re-established.

In *S. cerevisiae*, Rad51 promotes strand re-invasion, leading to a TS event during BIR (*Anand et al., 2014*). Presumably, Rad51 loads onto the ejected DNA strand following dissociation of the migrating D-loop to form a nucleoprotein filament. Both Srs2 and Fbh1 regulate nucleoprotein filament formation by dissociating Rad51 from DNA (*Krejci et al., 2003*; *Simandlova et al., 2013*; *Tsutsui et al., 2014*; *Veaute et al., 2003*). The loss of this anti-recombinogenic activity might account for the increase in TS observed in *srs2Δ* and *fbh1Δ* mutants.

Whilst dissociation of a nascent D-loop might prevent TS recombination, the unwinding of a fully established migrating D-loop would promote it. Indeed, this is exactly the activity that Mph1 is thought to catalyze during BIR in *S. cerevisiae* (*Stafa et al., 2014*). Surprisingly, we found that Fml1, despite promoting ectopic recombination during the restart of replication, appeared to have no effect on TS once replication had restarted. It is possible that a reduction in TS caused by lower levels of migrating D-loop dissociation is offset by an increased number of D-loops reaching our 12.4 kb genetic reporter. We are currently developing methods to measure restarted replication tract lengths, which will allow us to better calibrate our TS data.

In conclusion, we have identified a number of factors that influence TS associated with replication restart from a collapsed replication fork, revealing both similarities and potential differences with TS during BIR in *S. cerevisiae*. The fact that *S. pombe* and *S. cerevisiae* are only distant relatives, being about as divergent from each other as each is to animals (*Heckman et al., 2001*; *Sipiczki, 2000*), means that the similarities we have reported (e.g. Pif1 helicases promoting replication restart) are likely to be conserved in humans. Even the apparent disparities may simply reflect differences in the recombination assays used rather than fundamental mechanistic differences between RDR in *S.*

*pombe* and *S. cerevisiae*. This is exemplified in *S. cerevisiae*, where the importance of Mus81 in limiting TS during BIR appears to depend on the recombination assay (*Mayle et al., 2015*; *Stafa et al., 2014*). In future studies, it will be important to establish whether the apparent disparities between RDR in *S. pombe* and BIR in *S. cerevisiae* reflect real species-specific differences, fundamental differences in the restart of replication from a DSB/broken replication fork versus a collapsed, yet unbroken, fork or simply differences in the recombination assays that have been used.

# Materials and methods

**Key resources table**

| Reagent type (species) or resource | Designation | Source or reference | Identifiers | Additional information |
|---|---|---|---|---|
| Strain, strain background (*S. pombe*) | various strains | PMID: 15889146 | | standard laboratory strain (972) derivatives; see *Supplementary file 2* |
| Strain, strain background (*S. pombe*) | various strains | PMID: 25806683 | | standard laboratory strain (972) derivatives; see *Supplementary file 2* |
| Strain, strain background (*S. pombe*) | various strains | this paper | | standard laboratory strain (972) derivatives; see *Supplementary file 2* |
| Strain, strain background (*S. pombe*) | MCW4956 | PMID: 22426535 | | standard laboratory strain (972) derivatives; see *Supplementary file 2* |
| Strain, strain background (*S. pombe*) | various strains | PMID: 19546232 | | standard laboratory strain (972) derivatives; see *Supplementary file 2* |
| Strain, strain background (*S. pombe*) | MCW3059 | PMID: 18851838 | | standard laboratory strain (972) derivatives; see *Supplementary file 2* |
| Strain, strain background (*S. pombe*) | MCW3061 | PMID: 18851838 | | standard laboratory strain (972) derivatives; see *Supplementary file 2* |
| Recombinant DNA reagent | pMJ33 | this paper | | plasmid; see Materials and methods |
| Recombinant DNA reagent | pMJ34 | this paper | | plasmid; see Materials and methods |
| Recombinant DNA reagent | pCB44 | this paper | | plasmid; see Materials and methods |
| Recombinant DNA reagent | pMW899 | PMID: 22426535 | | plasmid |
| Recombinant DNA reagent | pMW905 | PMID: 22426535 | | plasmid |
| Sequence-based reagent | various oligonucleotides | this paper | | see *Supplementary file 3* (oligonucleotides) |

## Strains and plasmids

*S. pombe* strains and oligonucleotides are listed in *Supplementary files 2* and *3*, respectively. The construction of the '0 kb' recombination reporter, with *RTS1* positioned between the direct repeat

of *ade6*⁻ heteroalleles, has been described (*Ahn et al., 2005*). To integrate the *ade6*⁻ direct repeat recombination reporter ~35 kb, ~75 kb and ~140 kb downstream of the normal *ade6* locus, we used essentially the same strategy that was used previously to integrate the reporter at the 12.4 kb site (*Nguyen et al., 2015*), with the only difference being the construct used for targeted integration of *ade6-M375::kanMX6*. For targeted integration of *ade6-M375::kanMX6* at the 35 kb site we constructed pMJ33, which is a derivative of pMW923 (*Nguyen et al., 2015*) containing *ade6-M375:: kanMX6* flanked by DNA fragments amplified from genomic DNA using primers oMW1659 plus oMW1660 and oMW1661 plus oMW1662. pMJ34 and pCB44, which were used for targeted integration of *ade6-M375::kanMX6* at the 75 kb and 140 kb sites respectively, are similar to pMJ33 but contain genomic DNA fragments that were amplified using oMW1663 plus oMW1664 and oMW1665 plus oMW1666 (pMJ34) or oMW1621 plus oMW1622 and oMW1623 plus oMW1624 (pCB44). In each case, the targeting construct was liberated from its host plasmid by digestion with SpeI and PvuII. The strains with *tRNA^{GLU08}CD* or *tRNA^{GLU08}HO*, at the 12.4 kb *ade6*⁻ direct repeat recombination reporter site, were constructed in the same way as the standard 12.4 kb reporter strain using pMW899 or pMW905 instead of pFOX2 (*Nguyen et al., 2015*; *Steinacher et al., 2012*). Plasmids were verified by DNA sequencing and strains were verified by diagnostic PCR.

## Media and genetic methods

Protocols for the growth and genetic manipulation of *S. pombe*, and assays for recombination have been described (*Morrow et al., 2017*; *Nguyen et al., 2015*). Between 3 and 10 colonies were assayed in each recombination experiment, with experiments repeated at least three times to achieve a minimum sample size as calculated using the Power calculation $n = f(\alpha,\beta)(2\ s^2/\delta^2)$ where $\alpha = 0.05$; $\beta = 0.1$; $s = 40$; and $\delta = 50$. Strains being directly compared were analysed at the same time in parallel experiments. Statistical analysis of the recombination data was performed in SPSS Statistics Version 22 (IBM). Each data set was tested for normal distribution using a Shapiro-Wilk test, rejecting the null hypothesis (H₀; 'data fits a normal distribution') at an $\alpha$-level of $p < 0.05$. Several data sets did not conform to a normal distribution and, therefore, all comparisons were done using a two-tailed, two independent sample Wilcoxon rank-sum test (also known as the Mann-Whitney U test). This test is non-parametric and does not depend on data sets being normally distributed. Sample sizes and *p* values are given in *Supplementary file 1*.

## Two dimensional gel electrophoresis

Genomic DNA was prepared from asynchronously growing yeast cultures by enzymatic lysis of cells embedded in agarose and run on 2D gels as described (*Nguyen et al., 2015*). The ³²P-labelled probe for the 2D gel in *Figure 3* was made by random prime labeling using a template amplified from genomic DNA using primers oMW706 plus oMW707. The probe for the 2 D gel in *Figure 6— figure supplement 2* has been described (*Ahn et al., 2005*).

## Acknowledgements

We are grateful to Eleanor Hanson, Felix Kirsten and Kelly Lau for help in constructing and assaying some of the strains used in this study. We also thank Yota Murakami for the gift of the *pcf2Δ:: kanMX6* and *pcf3Δ::LEU2* strains.

## Additional information

### Funding

| Funder | Grant reference number | Author |
|---|---|---|
| Wellcome | 090767/Z/09/Z | Matthew C Whitby |
| Medical Research Council | MR/P028292/1 | Matthew C Whitby |
| Biotechnology and Biological Sciences Research Council | BB/P019706/1 | Matthew C Whitby |

The funders had no role in study design, data collection and interpretation, or the decision to submit the work for publication.

### Author contributions
Manisha Jalan, Conceptualization, Data curation, Formal analysis, Validation, Investigation, Visualization, Methodology, Writing—review and editing; Judith Oehler, Carl A Morrow, Formal analysis, Investigation, Writing—review and editing; Fekret Osman, Formal analysis, Supervision, Investigation, Methodology, Writing—review and editing; Matthew C Whitby, Conceptualization, Resources, Supervision, Funding acquisition, Visualization, Methodology, Writing—original draft, Project administration, Writing—review and editing

### Author ORCIDs
Manisha Jalan http://orcid.org/0000-0002-4467-4934
Judith Oehler http://orcid.org/0000-0002-8397-6492
Matthew C Whitby https://orcid.org/0000-0003-0951-3374

### Decision letter and Author response
Decision letter https://doi.org/10.7554/eLife.41697.020
Author response https://doi.org/10.7554/eLife.41697.021

## Additional files

### Supplementary files
• Supplementary file 1. Direct repeat recombinant frequencies.
DOI: https://doi.org/10.7554/eLife.41697.014

• Supplementary file 2. *Schizosaccharomyces pombe* strains.
DOI: https://doi.org/10.7554/eLife.41697.015

• Supplementary file 3. Oligonucleotides.
DOI: https://doi.org/10.7554/eLife.41697.016

• Transparent reporting form
DOI: https://doi.org/10.7554/eLife.41697.017

### Data availability
All data generated or analysed during this study are included in the manuscript and supporting files.

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
