## [Decision Letter]

Thank you for submitting your article "Factors affecting template switch recombination associated with restarted DNA replication" for consideration by *eLife*. Your article has been reviewed by two peer reviewers, one of whom is a member of our Board of Reviewing Editors, and the evaluation has been overseen by Detlef Weigel as the Senior Editor. The following individuals involved in review of your submission have agreed to reveal their identity: Gregory Ira (Reviewer #1).

The reviewers have discussed the reviews with one another and the Reviewing Editor has drafted this decision to help you prepare a revised submission.

Summary:

In this new manuscript the authors investigate the repair of stalled forks in fission yeast with a system where polar replication fork blockage – *RTS1* sequence – is placed downstream from a set of early firing origins. *RTS1* is known to stimulate recombination and the authors previously demonstrated that *RTS1* stimulates locally also template switches. The role of multiple enzymes in template switches at the site of the fork stalling is investigated. Template switches are scored as frequency of recombination between two Ade- alleles located downstream from *RTS1* that generates ADE+ colonies. The reporter cassette was placed at different distance from the fork stalling sequence and in some experiments the authors additionally modified the experimental system by deleting upstream origins, inserting a tRNA gene or replication that result in a delay of converging forks.

The authors provide convincing evidence suggesting that Pfh1 helicase is important for TS particularly further from the site of RTS1. This indicates that Pfh1 similar to Pif1 in budding yeast could be involved in repair specific extensive DNA synthesis. Furthermore the authors find a role of DNA helicases Srs2, Rqh1 and Fbh1 in controlling TS accordingly to previously documented functions of these helicases in regulation of recombination intermediates. Interestingly Fml1, a helicase that these authors and others showed to regulate D-loop stability has no function in limiting TS. Also Mus81 does not have a role in repair and TS frequency at stalled forks. The authors also show that TS can occur even far away from fork stalling place (75 kb) in case where there is no efficient converging fork. Overall the manuscript provides new and interesting data on a topic that is important in genome instability field.

The data are of high quality and the manuscript is well written. We note, however, that many of the effects in the mutants are small (and barely significant) and alternative models should be considered. Moreover, the effect of the tRNA gene on TS is not well connected to the rest of the manuscript.

Essential revisions:

1) You must reconsider how you discuss and compare your data with data obtained in other organisms. You should be more careful when comparing genetic requirements for template switches at nick induced recombination, in recombination at stalled replication forks and in BIR that occurs in G2. You make strong conclusions about the differences in control of TS between budding yeast and fission yeast but the differences in some of the phenotypes may be simply related to different recombination assays used and not different organisms as is claimed. Even within budding yeast depending on the recombination system use, the role of DNA helicases and nucleases in template switches and BIR is different. For example some BIR events in budding yeast depend on Mus81 while others do not and similarly TS are regulated by Mus81 or not depending on the assay. Stafa et al. showed that Mus81 is not regulating template switching during BIR while Mayle et al. showed that Mus81 suppresses template switching in recombination induced by nicks. Work from Russell and Arcangioli labs clearly demonstrated the function of Mus81 in repair of nick induced breaks in fission yeast. Others demonstrated similar function of Mus81 in budding yeast and other organisms. In the assay used here, all repair processes seem to be possible without any contribution of Mus81 so it is somewhat difficult to compare these results with assays where Mus81 is essential. Also Fml1, similar to budding yeast, Mph1 may play a role in TS in fission yeast but in BIR and not at stalled forks. A fair comparison between budding yeast and fission yeast would require related recombination assays. Thus all strong statements on differences between organisms should be toned down unless you compare similar recombination assays.

2) The tRNA gene data are not well connected to the rest of the manuscript. A better connection could be made by introducing the *pfh1, rqh1, srs2, fbh1, fml1, mus81, pcf2*, and *pcf3* mutants into this assay.

---

## [Author Response]

Essential revisions:1) You must reconsider how you discuss and compare their data with data obtained in other organisms. You should be more careful when comparing genetic requirements for template switches at nick induced recombination, in recombination at stalled replication forks and in BIR that occurs in G2. […] A fair comparison between budding yeast and fission yeast would require related recombination assays. Thus all strong statement on differences between organisms should be toned down unless you compare similar recombination assays.

We have toned down the Discussion by making the following changes:

“With this system, we have identified similarities with BIR in *S. cerevisiae*, as well as some potential differences.”

“In conclusion, we have identified a number of factors that influence TS associated with replication restart from a collapsed replication fork, revealing both similarities and potential differences with TS during BIR in *S. cerevisiae*. […] In future studies, it will be important to establish whether the apparent disparities between RDR in *S. pombe* and BIR in *S. cerevisiae* reflect real species-specific differences, fundamental differences in the restart of replication from a DSB/broken replication fork versus a collapsed, yet unbroken, fork or simply differences in the recombination assays that have been used.”

2) The tRNA gene data are not well connected to the rest of the manuscript. A better connection could be made by introducing the pfh1, rqh1, srs2, fbh1, fml1, mus81, pcf2, and pcf3 mutants into this assay.

To better integrate the tRNA gene data we have analysed the effect of *pfh1-m21* and *pfh1-mt** mutants on TS at the 12.4 kb reporter containing *tRNA*-HO. These new data are presented in a new version of Figure 4, which incorporates data from the original version of Figure 6 (note that in this re-organisation of the paper the original Figure 5 is now Figure 6). Our new data show that the *pfh1* mutants exhibit heightened levels of spontaneous recombination when *tRNA*-HO is present but, surprisingly, with *RTS1-AO* they exhibit the same fold increase in TS relative to wild-type as they do without *tRNA*-HO (compare Figure 2B and Figure 5D). These data suggest that Pfh1 is not required for limiting *tRNA*-HO-induced TS.